# Real-time tracking and prediction of COVID-19 infection using digital proxies of population mobility and mixing

Kathy Leung [1,2,3], Joseph T. Wu [1,2,3 ✉] & Gabriel M. Leung [1,2]

Digital proxies of human mobility and physical mixing have been used to monitor viral transmissibility and effectiveness of social distancing interventions in the ongoing COVID-19 pandemic. We develop a new framework that parameterizes disease transmission models with age-specific digital mobility data. By fitting the model to case data in Hong Kong, we are able to accurately track the local effective reproduction number of COVID-19 in near real time (i.e., no longer constrained by the delay of around 9 days between infection and reporting of cases) which is essential for quick assessment of the effectiveness of interventions on reducing transmissibility. Our findings show that accurate nowcast and forecast of COVID-19 epidemics can be obtained by integrating valid digital proxies of physical mixing into conventional epidemic models.

[1] WHO Collaborating Centre for Infectious Disease Epidemiology and Control, School of Public Health, LKS Faculty of Medicine, The University of Hong Kong, Hong Kong, SAR, China. [2] Laboratory of Data Discovery for Health (D24H), Hong Kong Science Park, Hong Kong, SAR, China. [3]These authors contributed equally: Kathy Leung, Joseph T. Wu. ✉email: joewu@hku.hk

Tracking the spread of COVID-19 infection in real time has been an elusive goal, given the necessary delay between infection and reporting. This delay consists of the incubation period (around 6 days), time between symptom onset and diagnosis (around 3 days), and the duration between confirmation and reporting (around half day)[1,2]. Therefore, there is around 9 days of delay even with instantaneous updating of case reports, assuming that testing is adequate to capture a consistent proportion of cases.

We previously estimated that possibly 44% of all COVID-19 infection events were pre-symptomatic transmission, i.e., during the last 2–3 days of the index-case incubation period; and 95% of all transmission would have taken place by day 5 after symptom onset[3]. Moreover, COVID-19, like SARS and MERS previously, show remarkable clustering and potential for superspreading events[4]. These are especially important if they take place in high-risk settings, such as nursing homes and institutional contexts. Taken together, it remains an urgent priority to develop new analytics that would allow truly real-time monitoring of transmissibility, thus the application of timely public health interventions in mitigation.

Digital proxies of human mobility and physical mixing have been shown to provide useful insights into disease transmission[5–11]. Specifically, during the ongoing COVID-19 pandemic, data of human mobility and mixing have been used to monitor the effectiveness of social distancing interventions[5,9,10]. Here, using COVID-19 in Hong Kong as an example, we describe a framework that integrates such digital proxies into conventional epidemic models to (i) track transmissibility in near real time; and (ii) generate nowcast and short-term forecast of the pandemic.

## Results

**Transmissibility of COVID-19 in Hong Kong**. The first imported and local case of COVID-19 in Hong Kong was confirmed on January 23, 2020, and January 30, 2020, respectively. In response to the pandemic, Hong Kong imposed progressively more restrictive interventions on inbound travelers thereafter. As such, the transmissibility of imported and local cases was expected to be inherently different and should be estimated separately, hence we stratified the local cases in Hong Kong into secondary cases of local and imported cases (based on their epidemiologic linkages documented in the COVID-19 surveillance database compiled by the Government Center for Health Protection) and estimated the transmissibility of local and imported cases over time (see "Methods" for details).

To estimate the transmissibility of COVID-19 in Hong Kong, we first approximated the epidemic curves by date of infection by deconvoluting the epidemic curves by date of onset with the incubation period[12–14]. We then applied the method described in Thompson et al.[15] to estimate the instantaneous effective reproduction number ($R_t$), which was defined as the mean number of secondary infections generated by a typically infectious case at time $t$. We call the resulting estimates of $R_t$ for local cases our empirical $R_t$ estimates. We used simulations to verify that this method for generating empirical $R_t$ estimates was generally accurate, except when the daily case count fell below ten (Supplementary Fig. 1).

The empirical $R_t$ estimate was around 2.5 when community transmission first began in mid-January. $R_t$ then dropped rapidly to around 1 during late-January after Wuhan was locked down on 23 January and stringent nonpharmaceutical interventions were implemented across all major cities in mainland China (Fig. 1). $R_t$ hovered around 1 in February, during which aggressive physical distancing measures, such as "work-from-home" among civil

servants as well as many other businesses, were implemented. Due to the relaxation of these measures and the importation of cases among more than 75,000 returnees from infected countries during the first week of March[16], $R_t$ rebounded sharply to mid-January levels (i.e., around 2.5) within a week. As the cases generated by such increase in transmission began to be registered by surveillance during the second week of March (i.e., around 9 days of infection-to-reporting delay), $R_t$ began to drop, probably due to spontaneous resumption of physical distancing measures among the general public in response to the observed increase in community transmission. NPIs such as "work-from-home" were reintroduced on 21 March and $R_t$ continued to drop thereafter to subcritical levels (i.e., <1) in April.

**Digital proxies for population mixing**. The basic premise of our framework was that intracity mobility and physical mixing relevant to the local spread of COVID-19 in Hong Kong could be gauged by the digital transactions made on Octopus cards, which are ubiquitously used by the Hong Kong population for their daily public transport and small retail payments (https://www.octopus.com.hk/tc/consumer/index.html). We demonstrated the validity of our premise by calculating the Pearson's correlation coefficients between these digital proxies and the posterior mean of the empirical $R_t$ estimates.

The empirical $R_t$ estimates were highly correlated with the number of Octopus transactions for transport: the Pearson's correlation coefficients between the two were $r = 0.62$, 0.68, 0.80, and 0.76 for children, students, adults, and the elderly, respectively (Fig. 2 and Supplementary Fig. 2). These results support our premise that Octopus transport transactions were valid digital proxies for population mixing. The correlation between Octopus retail transactions and the empirical $R_t$ estimates was low except for fast-food retail among adults ($r = 0.71$, which was still lower than that for adult transport). Consequently, we did not use retail transactions as digital proxies.

**Integrating the digital proxies into an epidemic model**. Assuming that Octopus transport transactions were valid digital proxies for population mixing, we integrated them into an age-structured (0–11, 12–18, 19–64, and ≥65 years) susceptible (S)-infectious (I)-removed (R) model, assuming that the contact matrix can be parameterized by optimally scaling the age-specific digital proxies (see "Methods"). We inferred these scaling factors and other model parameters by fitting the model to the epidemic curve of local cases between January 22, 2020 and May 31, 2020 (the date of symptom onset of the last case was 28 April, but this case was reported on 31 May and there were no cases reported between 15 and 31 May)[17,18].

The $R_t$ estimates from the fitted model (i.e., the dominate eigenvalue of the next-generation matrix in the fitted model at time $t$) had a significantly higher correlation with the empirical $R_t$ estimates than the constituent digital proxies ($r = 0.98$, Fig. 3). The deviations between the two in late-February (during which empirical $R_t$ was lower) and early-March (during which empirical $R_t$ was higher) might be due to low case counts (which tends to cause oscillations in the empirical $R_t$ estimates; see Supplementary Fig. 1). We estimated that only 23% (13–47%) of all local COVID-19 infections in Hong Kong had been ascertained by the official surveillance system (Supplementary Fig. 6). We performed the same model fitting at five earlier time points of the epidemic: 2 March, 14 March, 17 March, 22 March, and 4 April. The posterior distributions of the scaling factors that translate the digital proxies into the contact matrix were largely the same over the course of the epidemic (Supplementary Fig. 3). That is, the inferred mechanistic dependence of transmission dynamics on

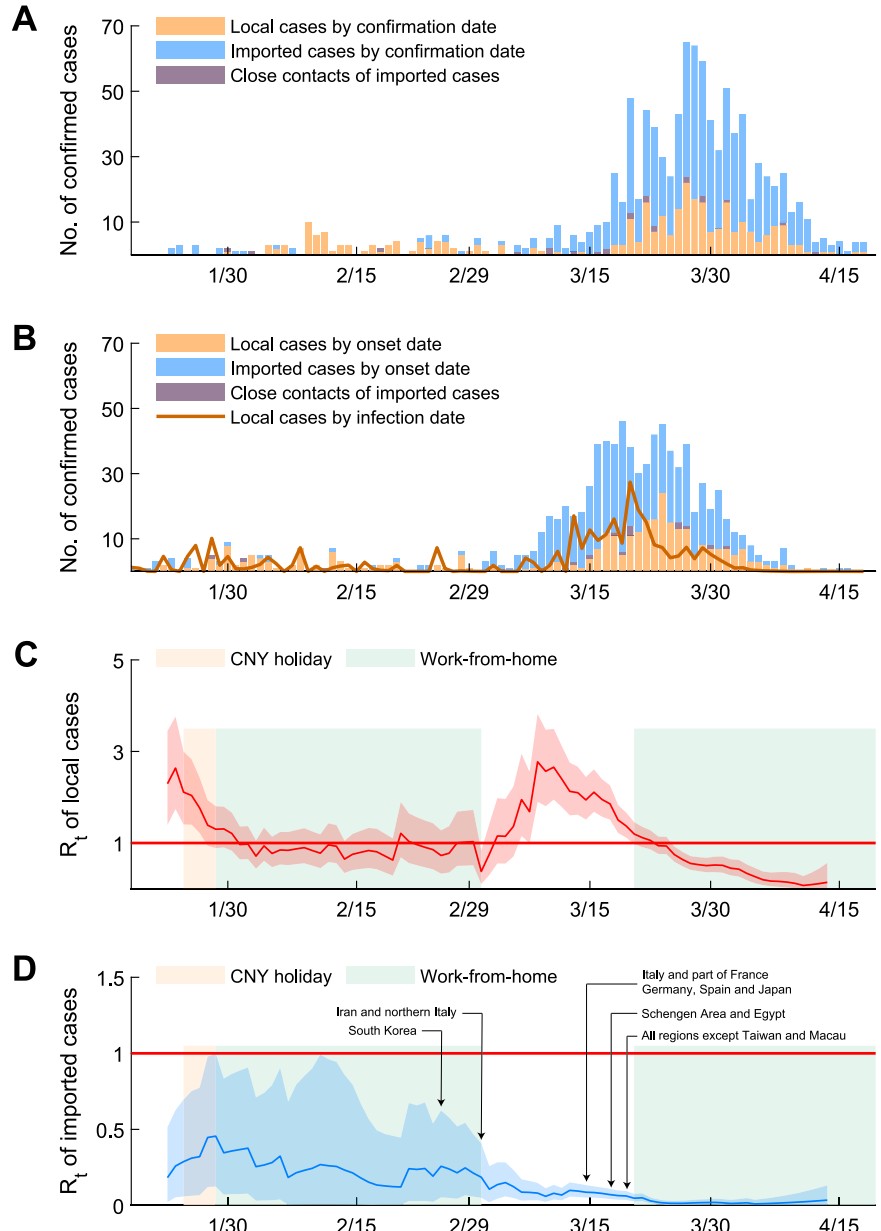

**Fig. 1 Transmissibility of COVID-19 in Hong Kong. A**, **B** The epidemic curves by dates of confirmation and symptom onset stratified by imported cases, close contacts of imported cases, and local cases. The epidemic curve of local cases by dates of infection was estimated using deconvolution. **C**, **D** $R_t$ estimates of local and imported cases by dates of infections on sliding weekly windows using methods from Thompson et al. Local cases only include CHP categories of local case, possibly local case, epidemiologically linked with local case and epidemiologically linked with the possible local case. The light shades showed the Chinese New Year holidays and the time period when "work-from-home" arrangements were implemented for civil servants. Fourteen-day quarantine was mandated for travelers from South Korea since 25 February; Iran and northern Italy since 1 March; Italy and affected area in France, Germany, Spain, and Japan since 14 March; Schengen area and Egypt since 17 March; all regions and countries except Taiwan and Macau since 19 March; and all regions and countries since 25 March. Red lines and shades indicated the posterior mean and 95% credible intervals of local $R_t$ estimates. Blue lines and shades indicated the posterior mean and 95% credible intervals of $R_t$ estimates for imported cases.

the digital proxies was stable over time, giving credence to the epidemiologic validity of our framework. Incorporating household contact patterns into our framework did not improve its performance (see Supplementary Information), probably because local cases have been generated mostly by community transmission.

**Nowcasting and forecasting the spread of COVID-19 in Hong Kong.** The traditional framework of generating empirical $R_t$

estimates from epidemic curves could not provide real-time estimates of $R_t$ because there was an inevitable delay of around 9 days between infection and case reporting on average (e.g., 6 days of incubation period plus 3 days of lead time between symptoms onset and case reporting). In contrast, our framework could be used to translate the digital proxies (which can be autonomously compiled by Octopus daily) into real-time estimates of $R_t$. More importantly, the fitted model could be used to (i) nowcast the epidemic (i.e., estimating the number of cases that have already been generated but not yet registered by surveillance

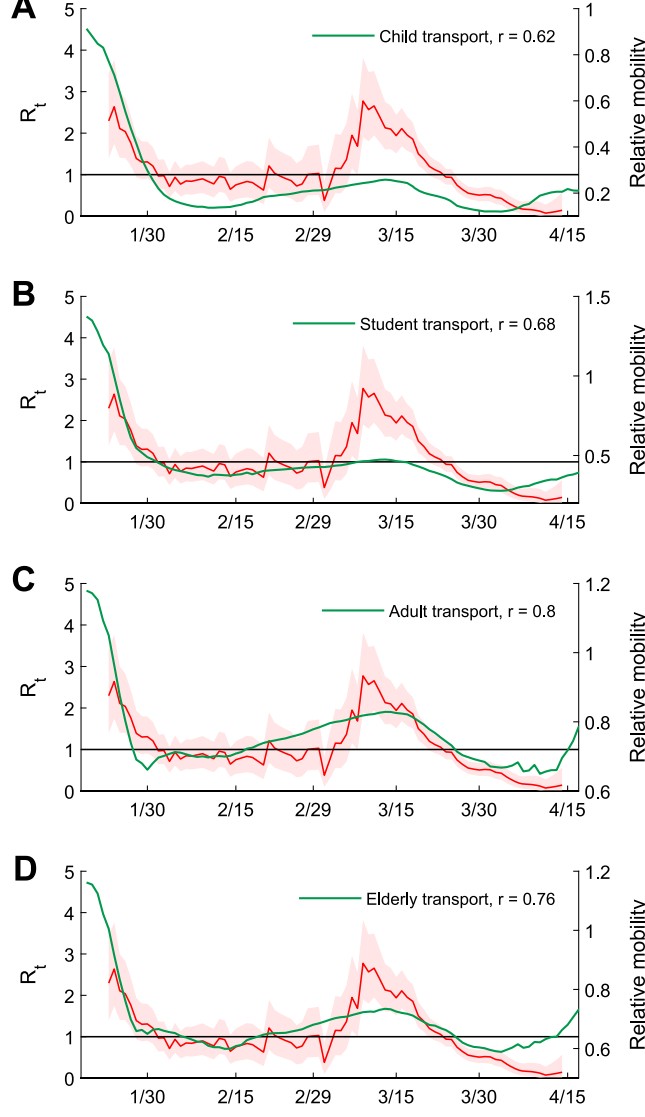

**Fig. 2 Correlation between transmissibility of COVID-19 and mobility levels inferred from Octopus transport data in Hong Kong. A–D** Correlation with Octopus transport data (7-day moving average) of children, students, adults, and the elderly. Red lines and shades indicated the posterior mean and 95% credible intervals of local $R_t$ estimates from Fig. 1.

due to the infection-to-reporting delay); and (ii) generate short-term epidemic forecast by making assumptions on how the digital proxies (i.e., population mixing) might evolve over the forecast time horizon.

We used the metrics proposed in Funk et al. (namely sharpness, bias, ranked probability score (RPS), Dawid–Sebastiani score (DSS), and absolute error (AE)) to assess the predictive performance of our forecasts (Supplementary Fig. 7)[19]. Sharpness, RPS, DSS, and AE dropped progressively between 30 January and 28 February, suggesting that the performance of our forecast model was improving as more mobility and case data became available (Supplementary Fig. 7). However, all metrics suggested that the forecasts were less accurate on 29 February, 15–17 March, and 22–23 March (Fig. 3 and Supplementary Fig. 8, in the sense that the observed case counts were near or outside the 95% prediction intervals). These prediction errors were likely due to the occurrence of superspreading events, which tend to result in more explosive growth in incidence[20], and the stochastic effects in an

otherwise low-prevalence setting. Specifically, our forecast underestimated the number of new onsets on 29 February which comprised a cluster from a religious group and another cluster seeded by returnees on the Diamond Princess Cruise. Similarly, our forecast underestimated incidence on 22–23 March which comprised a large cluster from a bar setting[17]. Because SSEs were rare in the training data and not explicitly modeled in our framework, the forecast model underestimated incidence when SSEs occurred. On the other hand, our forecast overestimated incidence on 15–17 March probably because the true prevalence was very low (on the order of tens) and the associated stochasticity in transmission dynamics resulted in an epidemic take-off that occurred slower than predicted by our deterministic framework. Notwithstanding the prediction errors attributed to SSEs and very low prevalence, our framework generated largely robust epidemic nowcasts during February–April: the estimated number of cases and its 95% prediction intervals from the fitted model (which were sufficiently tight for practical purposes) provided a reasonably robust inference of the number of cases that had already been generated but not yet registered by surveillance due to the infection-to-reporting delay (Fig. 3). Under the assumption that population mixing would remain at status quo for 6 days following the time of prediction, our epidemic forecasts were congruent with the observed epidemic curve during February–April except when SSEs occurred or when prevalence was very low (Fig. 3).

Finally, although the $R_t$ estimates and scaling factors for contact matrix parameterization were sensitive to assumptions regarding the generation time distribution (as expected), the accuracy and precision of the nowcasts and forecasts were unaffected (Supplementary Figs. 9 and 10).

## Discussion

Epidemic dynamics of directly transmitted infectious diseases, including COVID-19, is shaped by contact patterns which can fluctuate substantially over time due to spontaneous behavioral changes in physical mixing among the general public as well as interventions mandated by health officials (e.g., different components of "lockdowns")[21]. Although the conventional method of using social contact surveys to gauge contact patterns has provided valuable insights into the epidemiology of many infectious diseases (e.g., influenza, varicella, the current COVID-19 pandemic, etc.)[22–24], it might be difficult to acquire real-time updates of population mixing on a daily basis in the context of epidemic surveillance, especially in settings with limited resources. Mobile and location-based technology offer a complementary and efficient solution—the digital footprints of human mobility and activities registered by ubiquitously used mobile platforms (e.g., Octopus cards in Hong Kong, WeChat, and Alipay in mainland China, Oyster cards in the UK, Facebook and Google in the US, etc.) can be harnessed to generate near real-time proxies of population mixing with very high frequency and spatiotemporal resolution at very low cost. In this study, we have illustrated how accurate nowcast and short-term forecast of COVID-19 epidemics can be obtained using epidemic models parameterized with valid digital proxies even when population mixing was varying widely on a weekly or even daily basis.

The robustness of such a framework hinges on the identification of proxies that can provide representative and epidemiologically valid descriptions of human mobility and mixing among different age groups over time. The correlation between Octopus transactions and empirical $R_t$ estimates would be weaker if the former were not stratified by age and transaction categories (Supplementary Fig. 2). Other digital proxies for human mobility in Hong Kong such as CityMapper[25] and Google's community mobility reports[26] (Supplementary Fig. 4) had a lower correlation

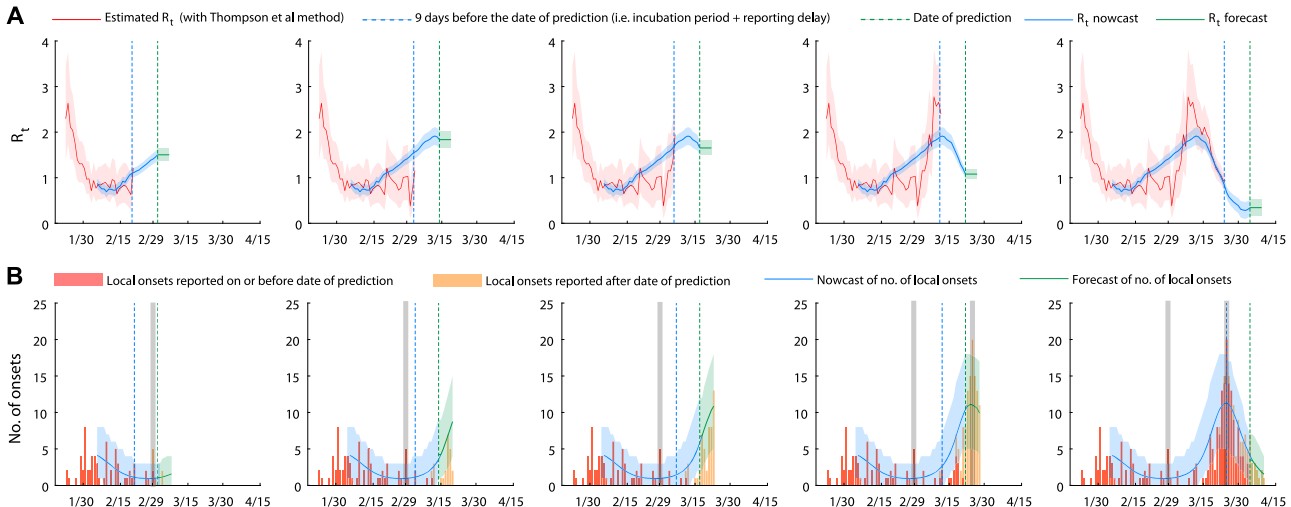

**Fig. 3 Retrospective nowcast and forecast of the COVID-19 epidemic in Hong Kong.** The green dash lines indicate the dates on which retrospective nowcast, and 6-day forecast were made (2 March, 14 March, 17 March, 22 March, and 4 April). The blue dash lines indicate the latest empirical Rt estimates obtainable from the epidemic curve on those dates. **A** Comparison between empirical $R_t$ estimates and $R_t$ from the fitted SIR model. The red line and shades indicate the empirical $R_t$ estimates (posterior mean and 95% credible intervals). Blue lines and shades indicate the nowcasted $R_t$ (posterior mean and 95% CrIs). Green lines and shades correspond to the assumption that population mixing (and hence $R_t$) would remain at status quo for the next 6 days (posterior mean and 95% CrIs). **B** Nowcast and forecast of local epidemic curves by dates of symptom onset. Red bars showed the number of local onsets reported on or before the date of prediction. Orange bars showed the number of local onsets reported after the date of prediction. Blue lines and shades.

with empirical $R_t$ estimates than Octopus transport transactions, probably because Octopus has much higher uptake across all age groups and areas. Similarly, we found that Baidu's transport data for Beijing, Shanghai, Shenzhen, and Wenzhou (the four Chinese cities for which empirical $R_t$ estimates have been reported in our previous study[27]) also correlated well with the respective local $R_t$ (Supplementary Fig. 5). In principle, jurisdictions that can track $R_t$ in near real-time would be able to adopt a reinforcement learning approach to optimize their intervention portfolios via rapid iterative cycles of transmissibility assessments and portfolio readjustments.

Our methods leveraged aggregate data instead of individual-level data from mobile phone usage. While individual-level data could probably provide more information to improve the temporal and spatial resolution of COVID-19 transmissibility, the use of personally identifiable information must be considered more thoughtfully to avoid the associated social, ethical and legal challenges[6]. Aggregate data of population mobility and activity, such as the Octopus card usage and Baidu's intracity traffic indices, are by default anonymized and therefore should not elicit privacy concerns.

Our current framework has several limitations. First, daily usage of Octopus cards among adults is much higher than that among children (e.g., public transportation is free for children aged below 3 years) and the elderly (e.g., the daily activities of many seniors tend to occur within walking distance of their neighborhood). As such, we posit that the explanatory and predictive power of our framework could be enhanced by including proxies that are more specific to young children as well as older adults (e.g., relative changes in customer volumes of "dim sum" restaurants that are regularly patronized by the elderly in Hong Kong).

Second, it is unlikely that big data such as Octopus transactions can reflect physical mixing within households. However, this seems to have little impact on how well our framework could nowcast and forecast the COVID-19 epidemic in Hong Kong, probably because community transmission has so far been the major driving force for local spread of COVID-19. Nonetheless,

our framework should be extended to deal with more general transmission scenarios by integrating the community contact patterns inferred from digital proxies with household contact patterns obtained from conventional social contact surveys.

Third, our framework tracked real-time changes in physical mixing but not temporal changes in the probability that these contacts conduce disease transmission. The latter might depend strongly on infection-prevention behaviors (e.g., wearing of masks and heightened personal hygiene) which can vary substantially as public sentiments regarding epidemic control fluctuate. Again, this seems to have little impact on the performance of our framework in this study, probably because infection-prevention precaution has consistently remained at very high levels among the general public in Hong Kong since the first emergence of COVID-19 in January 2020[16]. Future refinements of our framework should include tracking of factors that might affect transmissibility other than population mixing, which includes infection-prevention behaviors of the general public as well as climatic conditions[28].

Fourth, our framework is based on changes in population mixing and therefore might not be able to provide accurate epidemic forecast when there are not enough data from large clusters or superspreading events to train the model (Fig. 3). Changes in population mixing correlate with the average disease transmissibility well, but the occurrence and scale of superspreading events are often associated with substantial stochasticity. To improve our framework, estimation of the scale of superspreading events should be integrated into the model once the relevant data are available from outbreak investigations.

Fifth, our framework is based on deterministic epidemic dynamics and hence is not yet designed to cope with stochasticity associated with very low disease prevalence (e.g., at fadeout levels). To generate a more accurate nowcast and forecast in such epidemic settings, the disease transmission model and inference of its parameters in our framework should be extended to account for stochastic epidemic dynamics[29].

In conclusion, we have shown that digital proxies of population mobility and mixing can be integrated into conventional

epidemic models to nowcast and forecast epidemics. As the penetration of mobile and location-based technology continues to rise globally in the future, many jurisdictions would be able to harness various streams of real-time big data to generate timely and robust epidemic intelligence. Indeed, in the context of COVID-19 control, such data are already readily accessible in countries such as China (e.g., WeChat and Alipay transactions) for designing policies that maximize economic productivity while maintaining the effective reproductive number below 1 until safe and effective vaccines become widely accessible.

## Methods

**COVID-19 data in Hong Kong.** The daily number of confirmed COVID-19 cases in Hong Kong by date of symptoms onset was provided by the Centre for Health Protection (CHP). Patients who met certain clinical, epidemiological, or laboratory criteria were classified as suspected, probable, or confirmed cases in Hong Kong[2]. Although some SARS-CoV-2 infections were asymptomatic at the time of reporting, for simplicity we used the term "case" to include all symptomatic and asymptomatic SARS-CoV-2 infections. A confirmed case is defined as a patient whose specimens have been laboratory-confirmed to contain virologic or serologic evidence of infection with SARS-CoV-2, irrespective of symptoms or epidemiologic linkage. The testing strategy in Hong Kong has been relatively consistent between late-January and the end of April[2].

The CHP classified all COVID-19 cases into six categories: imported case, local case, possibly local case, epidemiologically linked with imported case, epidemiologically linked with local case and epidemiologically linked with possibly local case (https://www.coronavirus.gov.hk/eng/index.html). Transmissibility of imported and local cases was expected to be very different because intensive nonpharmaceutical interventions had been imposed on travelers arriving from COVID-19 affected regions since January 2020. Specifically, 14-day home quarantine has been mandated for all individuals arriving in Hong Kong from mainland China since 8 January, from South Korea since 25 February, from Iran and affected areas in Italy since 1 March, from Italy and affected regions in France, Germany, Japan, and Spain since 14 March, from the Schengen Area since 17 March and from all overseas countries and territories since 20 March[16].

**Octopus data in Hong Kong.** The basic premise of our framework was that intracity mobility and physical mixing relevant to the local spread of COVID-19 in Hong Kong could be gauged by the digital transactions made on Octopus cards, which are ubiquitously used by the Hong Kong population for their daily public transport and small retail payments (https://www.octopus.com.hk/tc/consumer/index.html). The Octopus cards are used by 99% of the population of Hong Kong aged 16 to 65 and the system handles more than 14 million transactions, worth over HK$180 million on a daily basis. (http://www.octopus.com.hk/en/corporate/about-octopus/profile/services/index.html).

We obtained the daily number of Octopus transactions in two major categories, namely transport and retail, among four types of cards, namely child (for children aged 3 and 11), student (for primary, secondary school, and university students <26 years old as well as student cardholders who receive discounted fares on selected routes), adult (for non-student adults aged under 65), and elder (for older adults aged 65 years or above). The daily numbers of Octopus transactions were normalized assuming the average number of Octopus transactions of each category of each age group between January 1, 2020 and January 15, 2020 was 100%.

**Model, inference, and analysis.** Our analysis comprised the following steps:

1. Estimate the instantaneous reproductive number $R_t$ of local cases in Hong Kong on each day $t$ between 22 January (day 0) and 15 May (day $T$) in the renewal equation model built in the "EpiEstim" package.
2. Correlate the resulting time series of empirical $R_t$ estimates (posterior means) with the daily volumes of different types of Octopus transactions. Select digital traces with high correlation coefficients with $R_t$ as mobility proxies for parameterizing the contact matrix of the SIR-type epidemic model in step 3.
3. On each day $t$, optimize the scaling factors that translate the digital proxies into the contact matrix by fitting the epidemic model to the epidemic curve between day 0 and $t$.
4. On each day $t$, use the fitted model from step 3 to (retrospectively) nowcast and forecast the number of new cases and compare with the corresponding realized epidemic curve.

We describe the details of each of these steps in the following subsections.

*Step 1: estimating the instantaneous reproductive number of COVID-19 in Hong Kong.* The instantaneous effective reproduction number $R_t$ is defined as the average number of secondary cases generated by cases on day $t$. If $R_t > 1$, the epidemic is expanding at time $t$, whereas $R_t < 1$ indicates that the epidemic size is shrinking at

time $t$. Since the epidemic curves provided by the CHP are based on the dates of symptom onset, we used a deconvolution-based method to reconstruct the COVID-19 epidemic curves by dates of infections[12–14]. Specifically, we used $f_{\text{incubation}}$, the probability density function (pdf) of the incubation period, to deconvolute the time series of the daily number of symptom onsets to reconstruct an epidemic curve of a daily number of new infections. We assumed the incubation period distribution was Gamma with a mean and SD of 6.5 and 2.6 days[1].

To account for the differential transmissibility between local and imported cases, we decomposed the epidemic curves by imported cases, local cases linked with imported cases, and local cases based on the CHP case classification: imported cases include only "imported cases" defined by CHP; local cases linked with imported cases include only cases "epidemiologically linked with imported case" defined by CHP; local cases include "local case", "possibly local case", cases "epidemiologically linked with the local case", and cases "epidemiologically linked with possibly local case" defined by CHP.

We then computed $R_t$ for imported and local cases separately from the respective epidemic curves using the "EpiEstim" R package developed by Thompson et al.[15]. We called the resulting estimates of $R_t$ for local cases our empirical $R_t$ estimates. We used the default prior in "EpiEstim" R package, i.e., assuming a prior distribution for the effective reproduction number with mean of 5 and SD of 5. We assumed that the generation time distribution was gamma with (i) mean 5.2 days and coefficient of variation 0.33 in the base case; and (ii) mean 4.2 and 6.2 days with the same coefficient of variation in the sensitivity analysis. We assumed a 7-day time window in $R_t$ estimation with "EpiEstim" because the Octopus transport volume during weekends was on average 60–70% that during weekdays.

*Step 2: select digital traces as proxies for population mobility and mixing in Hong Kong.* We obtained the number of Octopus transactions in two major categories, namely transport and retail, among four types of cards, namely child (for children aged 3 to 11), student (for primary, secondary school, and university students who receive discounted fares on selected routes; some university students are >18 years), adult (for non-student adults aged under 65), and elder (for older adults aged 65 years or above). We used $g_{a,c}(t)$ to denote the normalized number of transactions for card type $a$ and payment category $c$ on day $t$ (such that $g_{a,c}(t) = 1$ on 1 January 2020). We calculated the Pearson's correlation coefficient between each of these digital proxies $g_{a,c}(t)$, and the posterior mean of our empirical $R_t$ estimates. We found that all transport transactions exhibited a correlation coefficient of 0.5 or above with $R_t$ while retail transactions exhibited a much lower correlation. As such, we selected only the age-specific transport transactions as digital proxies for population mixing in the epidemic model in step 3. Specifically, we assumed that the number of infectious contacts between age group $a$ and $b$ (outside households) at time $t$ could be modeled as $\gamma_a g_{a,tran}(t)\,\gamma_b g_{b,tran}(t)$ where $\gamma_a > 0$ and $\gamma_b > 0$ were the scaling factors for the digital proxy of age group $a$ and $b$ (to be inferred in step 3), respectively. Under such formulation, the average rate at which an individual in age group $a$ made infectious contacts with age group $b$ at time $t$ was

$$\beta_{ab}(t) = \frac{\gamma_a g_{a,tran}(t)\gamma_b g_{b,tran}(t)}{N_a} \quad (1)$$

*Step 3: optimize the weights of the digital proxies by fitting the epidemic model to the observed number of confirmed cases on each day (retrospectively).* We used our previous age-structured SIR model to simulate the transmission of COVID-19 in Hong Kong[30]:

$$\frac{dS_a(t)}{dt} = -S_a(t)\pi_a(t) \quad (2)$$

$$\frac{\partial I_a(t,\tau)}{\partial t} + \frac{\partial I_a(t,\tau)}{\partial \tau} = -f_{\text{GT}}(\tau)I_a(t,\tau) \quad (3)$$

$$I_a(t,0) = S_a(t)\pi_a(t) \quad (4)$$

$$\frac{dR_a(t)}{dt} = \int_0^t f_{\text{GT}}(\tau)I_a(t,\tau)d\tau \quad (5)$$

$$N_a = S_a(t) + \int_0^t I_a(t,\tau)d\tau + R_a(t) \quad (6)$$

$$\pi_a(t) = \sum_{b=1}^m \int_0^t \frac{\beta_{ab}(t)}{N_b}I_b(t,\tau)d\tau \quad (7)$$

where

- $m$ was the number of age groups in the population.
- $S_a(t)$ and $R_a(t)$ were the numbers of susceptible and removed individuals in age group $a$ at time $t$.
- $I_a(t,\tau)$ was the number of infectious individuals in age group $a$ at time $t$ who were infected at time $t–\tau$.
- $N_a$ was the total number of people in age group $a$.

- $\pi_a(t)$ was the force of infection on age group $a$ at time $t$.
- $f_{GT}$ was the pdf of the generation time.

The time-varying next-generation matrix for this SIR model was:

$$\text{NGM}(t) = T_g \begin{bmatrix} \frac{\beta_{11}(t)S_1(t)}{N_1(t)} & \cdots & \frac{\beta_{1m}(t)S_1(t)}{N_1(t)} \\ \vdots & \ddots & \vdots \\ \frac{\beta_{m1}(t)S_m(t)}{N_m(t)} & \cdots & \frac{\beta_{mm}(t)S_m(t)}{N_m(t)} \end{bmatrix} \quad (8)$$

where $T_g$ was the mean generation time.

The effective reproduction number $R_t$ corresponded to the dominant eigenvalue of $\text{NGM}(t)$[31,32]. The incidence rate of infection and reported onsets in age group $a$ at time $t$ were calculated as follows:

$$A_{a,\text{infection}}(t) = S_a(t)\pi_a(t) \quad (9)$$

$$A_{a,\text{onset}}(t) = p_{\text{report}} \int_0^t A_{a,\text{infection}}(u) f_{\text{incubation}}(t-u) du \quad (10)$$

where $p_{\text{report}}$ was the proportion of infections ascertained by the CHP. We assumed that the epidemic was seeded by $M$ local infections on January 22, 2020.

The set of parameters that were subject to statistical inference, which we denoted by $\theta$, included: (i) the seed size $M$; (ii) the scaling factors $\gamma_a$'s; and (iii) the ascertainment proportion $p_{\text{report}}$. We estimated $\theta$ from the daily number of symptom onsets reported in Hong Kong (see Supplementary Table 1 for the list of inferred parameters) assuming that the observed number of cases was Poisson distributed with a mean equal to the number of cases in the fitted model. The likelihood function was

$$\prod_t \frac{(\lambda_t)^{n_t} e^{-n_t}}{n_t!} \quad (11)$$

where $\lambda_t = \sum_a A_{a,\text{onset}}(t)$ and $n_t$ were the expected (from the model) and observed number of reported onsets on day $t$. The statistical inference was performed in a Bayesian framework with noninformative (flat) priors using Markov Chain Monte Carlo. We used $P_t(\theta)$ to denote the posterior distribution of $\theta$ obtained by fitting the model to epidemic data up to day $t$. All analyses were conducted in MATLAB 2020a and R 4.0.0.

*Step 4: nowcast and forecast the daily number of confirmed cases using the fitted models.* On each day $t$, we drew 5000 samples of $\theta$ from $P_t(\theta)$ to parameterize the age-structured transmission model and simulated the number of new confirmed cases for day $t$ (i.e., nowcast) as well as day $t + 1, ..., t + 6$ (i.e., 6-day forecast) in each of these models. We then compared these simulated epidemic trajectories with the corresponding observed case counts in the CHP data to evaluate the predictive performance of the fitted model.

**Incorporating household contacts into the framework.** Conventionally, contact matrices for epidemic models of human-to-human transmissible respiratory diseases are constructed using data from social contact surveys[33]. Although we have previously conducted a social contact survey and estimated the contact matrix in Hong Kong (Supplementary Tables 2 and 3), the contact patterns outside households have likely changed substantially since then due to the COVID-19 pandemic as well as social unrest that has been ongoing since 2019[21]. Let $H_{ab}$ be the average number of household contacts that an individual in age group $a$ had in age group $b$ from our previous survey (Supplementary Table 3).

As discussed in the main text, the contact matrix parameterized with digital proxies largely corresponded to social contacts outside the households that drove community transmission. In our sensitivity analysis, we assumed that the relative contact pattern within the household was the same as that in our previous survey and incorporate the household contact matrix into our framework as follows:

$$\beta_{ab}(t) = \mu \frac{\gamma_a g_{a,tran}(t) \gamma_b g_{b,tran}(t)}{N_a} + (1-\mu) H_{ab} \quad (12)$$

where $\mu$ was a weight parameter subject to statistical inference.

*Empirical estimates of $R_t$ in mainland Chinese cities.* For Shenzhen and Wenzhou, dates of symptom onset were available for most cases. As in our analysis for Hong Kong, we used $f_{\text{incubation}}$, the probability density function (pdf) of the incubation period, to deconvolute the time series of the daily number of symptom onsets to reconstruct an epidemic curve of the daily number of new infections. For Beijing and Shanghai, dates of symptom onset of many cases were not available. Thus we used $f_{\text{infection-reporting}}$, the pdf of the time between infection and case reporting, to deconvolute the time series of the daily number of confirmed cases accordingly. We assumed $f_{\text{incubation}}$ and $f_{\text{onset-reporting}}$ were independent such that the pdf of the time between infection and reporting was

$$f_{\text{infection-reporting}}(t) = \int_0^t f_{\text{onset-reporting}}(t-u) f_{\text{incubation}}(u) du.$$ We assumed the distribution of the time between symptom onset and reporting was Gamma with

mean and standard deviation (SD) of 4.3 and 3.2 days, based on 186 cases reported in January–February 2020 in Beijing[27], which was consistent with the time between symptom onset and case reporting since February across China from the WHO-China Joint Mission Report[34].

In the vast majority of provinces and cities in mainland China, 14-day centralized quarantine had been mandated for individuals who had been to Hubei within 14 days from 23 January until late-April[27]. Cases in Beijing, Shanghai, Shenzhen, and Wenzhou were only categorized as imported and local cases. Consequently, we could not estimate $R_t$ of local and imported cases separately as we did in our analysis for Hong Kong. Instead, we estimated $R_t$ of local and imported cases in these cities by modifying the method by Thompson et al.[15] as follows.

For each city, let $I_t$ be the total number of new infections on day $t$ which comprised both local $\left(I_t^{\text{local}}\right)$ and imported $\left(I_t^{\text{imported}}\right)$ infections, i.e., $I_t = I_t^{\text{local}} + I_t^{\text{imported}}$. We assumed that the generation time distribution was the same for imported and local infectors. Let $\rho_t$ be the relative infectiousness of imported infections on day $t$ ($\rho_t \in [0,1]$) due to the 14-day mandatory quarantine on imported infections). Given $I_s^{\text{local}}$, $I_s^{\text{imported}}$ and $\rho_s$ for $s = 0, ..., t-1$, the expected number of incident local infections on day $t$ was

$$\begin{aligned} E\left(I_t^{\text{local}} | I_0^{\text{local}}, ..., I_{t-1}^{\text{local}}, I_0^{\text{imported}}, ..., I_{t-1}^{\text{imported}}, \rho_0, ..., \rho_{t-1}\right) \\ = R_t \sum_{s=1}^t \left(I_{t-s}^{\text{local}} + \rho_{t-s} I_{t-s}^{\text{imported}}\right) (F_{GT}(s) - F_{GT}(s-1)) \end{aligned} \quad (13)$$

where $F_{GT}$ was the cdf of the generation time. We then estimated $R_t$ using the same Bayesian method described in Thompson et al. under two extreme scenarios: (1) $\rho_t = 0$ for all time $t$ which implied imported infections did not generate any local infections; and (2) $\rho_t = 1$ for all time $t$ which implied imported infections were as infectious as local infections (Fig. 2B of Leung et al.[27]).

**Intracity mobility and activity indices of cities in mainland Chinese cities.** To estimate the intracity mobility levels of mainland Chinese cities, we obtained publicly available indices of intracity traffic volumes based on location-based services provided by Baidu (https://qianxi.baidu.com/). The index is a normalized ratio of a city's population with intracity movement within 24 h to a city's residential population, though the precise details of the normalization algorithm have not been made publicly available by Baidu on their website.

**Reporting summary**. Further information on research design is available in the Nature Research Reporting Summary linked to this article.

## Data availability

We collated epidemiological data from publicly available data sources (i.e., complete line list of all cases from websites of Centre for Health Protection Hong Kong: https://www.coronavirus.gov.hk/eng/index.html and https://data.gov.hk/en-data/dataset/hk-dh-chpsebcddr-novel-infectious-agent). All the epidemiological information that we used is available in the main text or the Supplementary Materials. The aggregate data of passenger numbers by card types (i.e., child, student, adult, and elder) were provided in the supplementary information. Other data, including the aggregate data of passenger numbers by public transportation means and aggregate data of transactions by retail categories, were provided by Octopus Cards Limited (Octopus). We have obtained consent from Octopus to share the aggregate data of transport transactions between January 1, 2020 and May 31, 2020. Our agreement with Octopus prohibits us from further sharing data with third parties but interested parties can contact Octopus to make the same data request. Data and codes used in the paper are available at https://github.com/kathyleung/Octopus_mobility_model.

## Code availability

Codes used in the paper are available at https://github.com/kathyleung/Octopus_mobility_model.

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

## Acknowledgements

We thank Octopus Cards Limited for providing aggregate data of passenger numbers by public transportation means and aggregate data of transactions by retail categories for the research. We thank Tim K. Tsang and Benjamin J. Cowling for useful discussion on estimating the effective reproduction number of local and imported cases. We thank Eric H.Y. Lau for the useful discussion on estimating the fraction of pre-symptomatic infections. We thank Di Liu, Miky Wong, and Chi-Kin Lam for technical support. This research was supported by a commissioned grant from the Health and Medical Research Fund (CID-HKU2) and General Research Fund (grant no.: 17110020) from the Government of the Hong Kong Special Administrative Region. The funding bodies had no role in study design, data collection and analysis, preparation of the paper, or the decision to publish.

## Author contributions

J.T.W., G.M.L., and K.L. designed the experiments. G.M.L. and K.L. obtained the Octopus data. K.L. collected the data. K.L. and J.T.W. analyzed the data. K.L., J.T.W., and G.M.L. interpreted the results and wrote the paper.

## Competing interests

The authors declare no competing interests.
