## [Peer Review File · Nature Communications]

REVIEWER COMMENTS

Reviewer #1 (Remarks to the Author):

In this paper, Leung and coauthors describe an age-structured modeling framework that incorporates digital proxies of human mobility from smart-card transactions to obtain accurate real-time estimates of the time-varying reproductive number in Hong Kong and use such estimates to produce short-term forecast of COVID-19 incidence in the country.

The manuscript is very well written and clear. I enjoyed reading it. In general, the results represent a significant advance in using digital proxies of human behavior to forecast the spread of COVID-19. Previous approaches have been looking for correlations between changes in mobility/mixing and changes in R_t only.

Here, the authors demonstrated quite convincingly that digital traces can be effectively integrated into an epidemic model to produce early estimates of R_t , in particular, earlier than it would be possible by considering the epidemic data that are inevitably affected by delays in reporting.

In conclusion, as a general assessment and in terms of importance, I feel the study has enough merits to be considered for publication in Nature Communications.

On a less positive note, I feel that not sharing the aggregated mobility data the study is based upon, strongly limits its replicability to other contexts and makes the study less impactful as it could be. I understand the legal matter behind this choice, but - as the authors state in the Discussion - aggregated data of this type should not elicit privacy concerns.

I commend the authors for sharing their code; I would invite them to find a way to share the aggregated mobility metrics as well.

As a more technical remark, I think the manuscript would benefit by extending the timeline under study beyond the date of April 15, maybe until mid June - if possible.

As Hong Kong has gone through a series of declining and resurgent R_t in the past months, it would be important to show that this approach is valid over a longer time span of the pandemic life-cycle.

Reviewer #2 (Remarks to the Author):

This manuscript aims to demonstrate the utility of using digital proxies of mobility in real-time estimation of transmissibility. The main finding is that data on transport use correlates with the time-varying reproduction number R_t , and that therefore the delays inherent in real-time estimation of R_t can be eliminated. This is an important question and would be a finding of considerable significance.

Overall, my two main concerns about the paper: 1) the methodology is not described in sufficient detail, 2) the code provided was not sufficient to reproduce results and 3) the main finding is not convincingly demonstrated.

1) I found it really hard to understand what exactly had been done and why. Methodology is described in three places: the Methods section, the Supplementary Material, and the beginning of the Results section. I would suggest to synthesise this in the Methods section and clearly state the relationship between the different models used (from what I understand, one backcalculation model, one renewal equation model, and one SIR-type model) and the role of each of the models in testing the hypothesis posed.

2) The code sent with the paper seemed to be incomplete (only contained Matlab, no R code) and undocumented. I found it difficult where to start in reviewing and reproducing this, and it seemed to be missing files even beyond the private Octopus data. Also, I do not know what journal policy is but to me "The codes (without Octopus data) are available upon request to the corresponding author" is not an acceptable code availability statement. I would also strongly suggest to include code with the paper or make it public to allow for replication of the study.

3) The authors make a specific claim about the ability to get a more timely estimate of R_t from mobility data. To address this, Pearson's correlation coefficients are first calculated. I worry about how they might perform if the underlying data is far from uniformly distributed (i.e., the estimates could be biased by outliers), which I suspect might be the case here. I would suggest that other correlation coefficients (e.g., Spearman's) be considered and a scatter plot provided. The second part of the analysis (nowcasts/forecasts) more specifically addresses the main claim, but all that is presented is a plot, from which the reader is supposed to conclude that the method "accurately predicted the number of cases that had already been generated but not yet registered by surveillance due to the infection-to-reporting delay". I think that this needs an appropriate statistical analysis that demonstrates that the method yields accurate forecasts (e.g., MAE, RMSE or, even better, a proper scoring rule) and an improvement over a method that does not include digital proxies.

More specific comments

Main text

--

l.25: ref 1 is about the incubation period - the other delays in Hong Kong should be referenced, too

l.37: I do not see how ref 4 shows how "data of human mobility and mixing have been used to monitor viral transmissibility" - this needs at least one other reference

l.56/57: Figure S1 does not demonstrate accuracy. This would require some numerical analysis. Some of the estimates appear off (e.g., top row, or central panel). Also, I think the reliability of the method depends on the ability to estimate R_t separately from imported and local cases, which is not shown here.

l.76: a Pearson's correlation coefficient was calculated - of what? R_t has uncertainty, how was the corresponding distribution turned into a single number at each time point?

l.77: apart from providing another measure of correlation, it would be useful to see this estimated over different time periods. I can see that there appears to be good correlation early on but this seems to break down later, presumably because people were encouraged to resume economic activity (i.e., be mobile), whilst infection control measures were in place to keep $R_t < 1$.

l.115: "it accurately predicted the number of cases" is a strong claim that is central to the main message to the paper. This needs a full statistical analysis.

l.133: the authors claim that "the conventional method of using social contact surveys ... is too time- and resource-intensive for acquiring real-time updates of population mixing in the context of epidemic surveillance" but this is being done now in many countries, e.g.

<https://www.medrxiv.org/content/10.1101/2020.04.13.20064014v2> and

<https://bmcmmedicine.biomedcentral.com/articles/10.1186/s12916-020-01597-8>

l.152: the Discussion is not an appropriate place to introduce a new bit of analysis in the supplement, based on data that has not been described; the data sources and analysis should be described somewhere and referenced in the Results

l.200: the Methods are missing a description of the data - are they confirmed cases, how confirmed? Was testing done based on symptoms, and was the same testing strategy applied over the time period considered?

l.212: seems to sum over the wrong index.

l.220: sources/references should be given for all the publicly available data source and, if they are web sources, the data included with the paper in case the sources disappear.

I.252: could the estimated infection curve be shown?

Supplement

--

I.15/16: "we used a deconvolution-based method" needs more detail and equations and/or code

I.21: mean and SD have uncertainty in ref. 3 - what is the rationale for ignoring this, and what are the consequences on results? There seems to be a risk of spurious precision from ignoring this

I.26: is the Gamma distribution a good choice given that the times are discrete? How was it converted into discrete counts of days?

I.28: presumably the issue of missing uncertainty is also affecting the estimates of the delay from onset to reporting

I.33: when using the Thompson et al. method, what prior was used, what generation time interval distribution and what time window, and why?

I.49: the model equations are missing some definitions, e.g. what is π_a , what is $I_b(t, \tau)$? All symbols used should be introduced and not left for the reader to guess

I.53/54: "pdf of the generation time was assumed to be Gamma and the same as that of the serial interval" - this is known to be wrong (see, e.g., Ganyani et al., 2020); what are the consequences of this assumption?

I.73/74: also seems to confuse the serial and generation interval which, I think, are important to separate even if assumed equal

I.109: where do asymptomatic infections feature and are they assumed to have the same generation interval as symptomatic cases on which the estimates were based? Also, is there a chance that the generation interval changed over the course of the epidemic (e.g., due to control measures) and what are the consequences of this?

Table S2/S3: are these missing uncertainty? Is this uncertainty reflected in the model?

Fig S1 is hard to assess visually - perhaps compare the R_t estimates with a scatter plot? R_t estimates are shown with uncertainty but since this is turned into a single value for the Pearson's correlation coefficient, would it make sense to just show this aggregate value here?

Fig. S3: the posterior distributions look reassuringly similar but it would be good to also show parameter traces to assure the reader that convergence was obtained in the MCMC

Point-by-point response to comments for “Real-time tracking and prediction of COVID-19 infection using digital proxies of population mobility and mixing”

Reviewer #1 (Remarks to the Author):

Comment 1.1. In this paper, Leung and co-authors describe an age-structured modelling framework that incorporates digital proxies of human mobility from smart-card transactions to obtain accurate real-time estimates of the time-varying reproductive number in Hong Kong and use such estimates to produce short-term forecast of COVID-19 incidence in the country.

The manuscript is very well written and clear. I enjoyed reading it. In general, the results represent a significant advance in using digital proxies of human behavior to forecast the spread of COVID-19. Previous approaches have been looking for correlations between changes in mobility/mixing and changes in R_t only. Here, the authors demonstrated quite convincingly that digital traces can be effectively integrated into an epidemic model to produce early estimates of R_t , in particular, earlier than it would be possible by considering the epidemic data that are inevitably affected by delays in reporting.

In conclusion, as a general assessment and in terms of importance, I feel the study has enough merits to be considered for publication in Nature Communications.

Response 1.1. Thank you for the supportive comments. Please see below for our point-by-point response.

Comment 1.2. On a less positive note, I feel that not sharing the aggregated mobility data the study is based upon, strongly limits its replicability to other contexts and makes the study less impactful as it could be. I understand the legal matter behind this choice, but - as the authors state in the Discussion - aggregated data of this type should not elicit privacy concerns. I commend the authors for sharing their code; I would invite them to find a way to share the aggregated mobility metrics as well.

Response 1.2. Thank you for your comment. We have obtained consent from the Octopus Cards Limited (Octopus) to share the aggregate data of transport transactions by card types (i.e. child, student, adult, and elder) between 1 January and 31 May 2020. We have revised the “**Data availability**” section in the manuscript accordingly.

Comment 1.3. As a more technical remark, I think the manuscript would benefit by extending the timeline under study beyond the date of April 15, maybe until mid-June - if possible. As Hong Kong has gone through a series of declining and resurgent R_t in the past months, it would be important to show that this approach is valid over a longer time span of the pandemic life-cycle.

Response 1.3. Thank you for your comment. As discussed in our response to **Comment 1.2.**, we have obtained consent from Octopus Cards Limited (Octopus) to share the aggregate data of transport transactions by card types (i.e. child, student, adult, and elder) between 1 January and 31 May 2020.

Reviewer #2 (Remarks to the Author):

Comment 2.1. This manuscript aims to demonstrate the utility of using digital proxies of mobility in real-time estimation of transmissibility. The main finding is that data on transport use correlates with the time-varying reproduction number R_t , and that therefore the delays inherent in real-time estimation of R_t can be eliminated. This is an important question and would be a finding of considerable significance. Overall, my two main concerns about the paper: 1) the methodology is not described in sufficient detail, 2) the code provided was not sufficient to reproduce results and 3) the main finding is not convincingly demonstrated.

Response 2.1. Thank you for your comments. We have substantially revised the manuscript and the supplementary codes and data. Please see below for our point-by-point response.

Comment 2.2. 1) I found it really hard to understand what exactly had been done and why. Methodology is described in three places: the Methods section, the Supplementary Material, and the beginning of the Results section. I would suggest to synthesise this in the Methods section and clearly state the relationship between the different models used (from what I understand, one backcalculation model, one renewal equation model, and one SIR-type model) and the role of each of the models in testing the hypothesis posed.

Response 2.2. Thank you for your comment and we apologize for the confusion. Our results were presented in the main text with the order as follows:

- 1) **“Transmissibility of COVID-19 in Hong Kong”**: we estimated the effective reproductive number R_t for both imported and local cases in Hong Kong using the “EpiEstim” package
- 2) **“Digital proxies for population mixing”**: we calculated Pearson’s correlation coefficients between the posterior mean of R_t estimates from “EpiEstim” and various digital proxies of mobility and mixing obtained from Octopus transactions (including transport and retail transactions from different card types, Figure S2). We used these resulting Pearson’s correlation coefficients to identify the most suitable candidate Octopus proxies of mobility and mixing.
- 3) **“Integrating the digital proxies into an epidemic model”**: we then integrated the candidate Octopus proxies into building an SIR-type epidemic model and fitted the model to data of local cases again. Parameters as shown in Table S1 were estimated from the inference.
- 4) **“Nowcasting and forecasting the spread of COVID-19 in Hong Kong”**: Using the fitted model and the posterior distribution of the parameters, we provided nowcast and forecast of the number of

local cases which should have been generated/infected but not yet registered in the surveillance database due to the infection-to-reporting delay.

Thus, we have only two models in the manuscript: 1) the model used by the “EpiEstim” package; and 2) the SIR-type model we built by incorporating the digital proxies into the construction of the next generation matrix. We did not include any renewal equation model throughout the analysis. We have also revised the Methods section accordingly to clarify, with the following steps:

- 1) **Step 1: Estimating the instantaneous reproductive number of COVID-19 in Hong Kong**
- 2) **Step 2: Select digital traces as proxies for population mobility and mixing in Hong Kong**
- 3) **Step 3: Optimize the weights of the digital proxies by fitting the epidemic model to the observed number of confirmed cases on each day (retrospectively)**
- 4) **Step 4: Nowcast and forecast the daily number of confirmed cases using the fitted models**

Please see the main text and Methods section for details.

Comment 2.3. 2) The code sent with the paper seemed to be incomplete (only contained Matlab, no R code) and undocumented. I found it difficult where to start in reviewing and reproducing this, and it seemed to be missing files even beyond the private Octopus data. Also, I do not know what journal policy is but to me "The codes (without Octopus data) are available upon request to the corresponding author" is not an acceptable code availability statement. I would also strongly suggest to include code with the paper or make it public to allow for replication of the study.

Response 2.3. We have obtained consent from the Octopus Cards Limited (Octopus) to share the aggregate data of transport transactions by card types (i.e. child, student, adult, and elder) between 1 January and 31 May 2020. Therefore, we have also revised the codes (with documentation) and uploaded them again as part of the supplementary information. We have revised the “**Data availability**” and “**Code availability**” sections in the manuscript accordingly.

Comment 2.4. 3) The authors make a specific claim about the ability to get a more timely estimate of R_t from mobility data. To address this, Pearson’s correlation coefficients are first calculated. I worry about how they might perform if the underlying data is far from uniformly distributed (i.e., the estimates could be biased by outliers), which I suspect might be the case here. I would suggest that other correlation coefficients (e.g., Spearman's) be considered and a scatter plot provided. The second part of the analysis (nowcasts/forecasts) more specifically addresses the main claim, but all that is presented is a plot, from which the reader is supposed to conclude that the method "accurately predicted the number of

cases that had already been generated but not yet registered by surveillance due to the infection-to-reporting delay". I think that this needs an appropriate statistical analysis that demonstrates that the method yields accurate forecasts (e.g., MAE, RMSE or, even better, a proper scoring rule) and an improvement over a method that does not include digital proxies.

Response 2.4. We did not attempt to get a more timely R_t estimate by calculating the Pearson's correlation coefficients between Octopus mobility proxies and the R_t estimates obtained by using "EpiEstim". Instead, we only calculated the Pearson's correlation coefficients between the digital proxies and the posterior mean of the R_t estimates and used the Pearson's correlation coefficients to identify the most appropriate Octopus transaction proxies to build our own model (Please see our response to **Comment 2.2**). Therefore, we did not attempt to explore a more appropriate measure for the correlation between digital proxies and the posterior distribution of R_t estimates, because the subsequent model building, epidemic nowcast and forecast did not involve the Pearson's correlation coefficients.

We presented the findings of the modelled nowcasts and forecasts in Figure 3, using a standard approach to show the estimated number of cases (blue and green lines in Figure 3B) and the 95% CrI (blue and green shades in Figure 3B) generated by the fitted model and the corresponding posterior distribution of parameters. The statistical measures proposed by the reviewer might not be appropriate in this context because a Poisson likelihood was used to fit the model to the number of reported cases and the stochastic effects would be large when the number of reported cases were lower than 10.

More specific comments

Main text

Comment 2.5. 1.25: ref 1 is about the incubation period - the other delays in Hong Kong should be referenced, too

Response 2.5. We have referenced the below preprint manuscript for the onset-to-report delays:
Wu P, Tsang TK, Wong JY, Ng TW, Ho F, Gao H, Adam DC, Cheung DH, Lau EH, Lim WW, Ali ST. Suppressing COVID-19 transmission in Hong Kong: an observational study of the first four months.
<https://www.researchsquare.com/article/rs-34047/v1>

Comment 2.6. 1.37: I do not see how ref 4 shows how "data of human mobility and mixing have been used to monitor viral transmissibility" - this needs at least one other reference

Response 2.6. We apologize for the confusion. We did not mean data of mobility and mixing have been used to monitor viral transmissibility but was trying to show data of mobility and mixing have been used to monitor the effectiveness of social distancing interventions. We have revised the sentence and added one more reference as below:

“Specifically, during the ongoing COVID-19 pandemic, data of human mobility and mixing have been used to monitor the effectiveness of social distancing interventions.”

Kishore N, Kiang MV, Engø-Monsen K, Vembar N, Schroeder A, Balsari S, Buckee CO. Measuring mobility to monitor travel and physical distancing interventions: a common framework for mobile phone data analysis. *The Lancet Digital Health*. 2020 Sep 1.

Comment 2.7. 1.56/57: Figure S1 does not demonstrate accuracy. This would require some numerical analysis. Some of the estimates appear off (e.g., top row, or central panel). Also, I think the reliability of the method depends on the ability to estimate R_t separately from imported and local cases, which is not shown here.

Response 2.7. We agree with the reviewer that our methods rely on the ability to estimate R_t separately while a substantial proportion of reported cases were imported cases and the level of complete information the surveillance system could provide for the source of infection of each reported case. We used the official categorization of imported and local cases which are available on the website of Hong Kong’s Centre for Health Protection: https://www.chp.gov.hk/files/pdf/local_situation_covid19_en.pdf

But our methods remain valid once the transmission is dominated by local transmission. Thus, we just used Figure S1 to show how accurate R_t could be estimated from deconvoluted epidemic curves of reported local cases. As pointed out by the reviewer, the estimates of R_t were quite accurate except in the scenarios when daily case count was below 10 (i.e. top row or central panel when the case count fell below 10 before Day 40 for $p_{report} = 0.05$ and before Day 20 for $p_{report} = 0.1$).

Comment 2.8. 1.76: a Pearson's correlation coefficient was calculated - of what? R_t has uncertainty, how was the corresponding distribution turned into a single number at each time point?

Response 2.8. We apologize for the confusion. We calculated the Pearson’s correlation coefficient between the digital proxies and the posterior mean of the empirical R_t estimates (obtained by using “EpiEstim” package and the methods by Thompson et al, *Epidemics*, 2019: <https://www.sciencedirect.com/science/article/pii/S1755436519300350>). We only used the Pearson’s correlation coefficients to identify appropriate Octopus transaction proxies to build our own model

(Please see our response to **Comment 2.2**). We did not attempt to explore a more appropriate measure for the correlation between digital proxies and the posterior distribution of R_t estimates, because the subsequent model building, epidemic nowcast and forecast did not involve the Pearson's correlation coefficients. We revised the sentence as follows for greater clarity:

“We demonstrated the validity of our premise by calculating the Pearson's correlation coefficients between these digital proxies and the posterior mean of the empirical R_t estimates.”

Comment 2.9. 1.77: apart from providing another measure of correlation, it would be useful to see this estimated over different time periods. I can see that there appears to be good correlation early on but this seems to break down later, presumably because people were encouraged to resume economic activity (i.e., be mobile), whilst infection control measures were in place to keep $R_t < 1$.

Response 2.9. As discussed in **Response 2.2. and 2.8.**, the Pearson's correlation coefficients were only used to identify appropriate Octopus transaction proxies to build the model. Figure S2 showed that Octopus transport (i.e. the total transport including MTR, buses, minibuses and other transports) outperformed other categories of Octopus transactions. We did not attempt to explore a more appropriate measure for the correlation between digital proxies and the posterior distribution of R_t estimates, because the subsequent model building, epidemic nowcast and forecast did not involve the Pearson's correlation coefficients.

Comment 2.10. 1.115: "it accurately predicted the number of cases" is a strong claim that is central to the main message to the paper. This needs a full statistical analysis.

Response 2.10. As discussed in our response to **Comment 2.4.**, we presented the findings of model nowcasts and forecasts in Figure 3, which used a standard approach to show the estimated number of cases (blue and green lines in Figure 3B) and the 95% CrI (blue and green shades in Figure 3B) generated by the fitted model and the corresponding posterior distribution of parameters. We have revised the manuscript accordingly to clarify:

“Our framework generated robust epidemic nowcasts during February-April: the estimated number of cases and its 95% CrI from the fitted model (which were reasonably tight for practical purposes) provided an accurate prediction of the number of cases that had already been generated but not yet registered by surveillance due to the infection-to-reporting delay (Figure 3). Under the assumption that population mixing would remain at status quo for 6 days following the time of prediction, all of the epidemic

forecasts from our framework converged with the observed epidemic curve during February-April (Figure 3).”

Comment 2.11. 1.133: the authors claim that "the conventional method of using social contact surveys ... is too time- and resource-intensive for acquiring real-time updates of population mixing in the context of epidemic surveillance" but this is being done now in many countries, e.g.

<https://www.medrxiv.org/content/10.1101/2020.04.13.20064014v2> and

<https://bmcmmedicine.biomedcentral.com/articles/10.1186/s12916-020-01597-8>

Response 2.11. By using “real-time” we meant the contact matrices can be updated on a daily basis with updated mobility data. We revised the sentence in the Discussion as follows and added the two references kindly provided by the reviewer:

“Although the conventional method of using social contact surveys to gauge contact patterns has provided valuable insights into the epidemiology of many infectious diseases (e.g. influenza, varicella, the current COVID-19 pandemic, etc.), it might be difficult to acquire real-time updates of population mixing on a daily basis in the context of epidemic surveillance, especially in settings with limited resources.”

Comment 2.12. 1.152: the Discussion is not an appropriate place to introduce a new bit of analysis in the supplement, based on data that has not been described; the data sources and analysis should be described somewhere and referenced in the Results

Response 2.12. We apologize for the confusion. The analysis in Figure S5 was not “new” but in fact it was part of our previous work on R_t estimation in mainland China. We have revised the sentence in the Discussion to minimize confusion:

“Similarly, we found that Baidu’s transport data for Beijing, Shanghai, Shenzhen and Wenzhou (the four Chinese cities for which empirical R_t estimates have been reported in our previous study) also correlated well with the respective local R_t (Figure S5). In principle, jurisdictions that can track R_t in near real-time would be able to adopt a reinforcement learning approach to optimize their intervention portfolios via rapid iterative cycles of transmissibility assessments and portfolio readjustments.”

It was from the supplementary appendix of the below paper:

Leung K, Wu JT, Liu D, Leung GM. First-wave COVID-19 transmissibility and severity in China outside Hubei after control measures, and second-wave scenario planning: a modelling impact assessment. The Lancet. 2020 Apr 8.

We have revised the Methods section and supplementary information accordingly.

Comment 2.13. 1.200: The Methods are missing a description of the data - are they confirmed cases, how confirmed? Was testing done based on symptoms and was the same testing strategy applied over the time period considered?

Response 2.13. Thanks for the comment. We have included the description of data in the Methods section now as follows:

“Patients who met certain clinical, epidemiological or laboratory criteria were classified as suspected, probable or confirmed cases in Hong Kong ². Although some SARS-CoV-2 infections were asymptomatic at the time of reporting, for simplicity we used the term “case” to include all symptomatic and asymptomatic SARS-CoV-2 infections. A confirmed case is defined as a patient whose specimens have been laboratory-confirmed to contain virologic or serologic evidence of infection with SARS-CoV-2, irrespective of symptoms or epidemiologic linkage. The testing strategy in Hong Kong has been relatively consistent between late January and the end of April ².

The CHP classified all COVID-19 cases into six categories: imported case, local case, possibly local case, epidemiologically linked with imported case, epidemiologically linked with local case and epidemiologically linked with possibly local case (<https://www.coronavirus.gov.hk/eng/index.html>). Transmissibility of imported and local cases were expected to be very different because intensive non-pharmaceutical interventions had been imposed on travelers arriving from COVID-19 affected regions since January 2020. Specifically, 14-day home quarantine has been mandated for all individuals arriving in Hong Kong from mainland China since 8 January, from South Korea since 25 February, from Iran and affected areas in Italy since 1 March, from Italy and affected regions in France, Germany, Japan and Spain since 14 March, from the Schengen Area since 17 March and from all overseas countries and territories since 20 March ¹⁵.”

“To account for the differential transmissibility between local and imported cases, we decomposed the epidemic curves by imported cases, local cases linked with imported cases, and local cases based on the CHP case classification: imported cases include only “imported cases” defined by CHP; local cases linked with imported cases include only cases “epidemiologically linked with imported case” defined by CHP; local cases include “local case”, “possibly local case”, cases “epidemiologically linked with local case” and cases “epidemiologically linked with possibly local case” defined by CHP.”

Comment 2.14. 1.212: seems to sum over the wrong index.

Response 2.14. We apologize for the confusion. Here we used n_a to denote the number of candidate digital proxies of the age group a . We have added the explanation in the Methods section.

Comment 2.15. 1.220: sources/references should be given for all the publicly available data source and, if they are web sources, the data included with the paper in case the sources disappear.

Response 2.15. We have provided the web links for the case data which are permanently available on websites hosted by the Hong Kong government:

<https://www.coronavirus.gov.hk/eng/index.html>

<https://data.gov.hk/en-data/dataset/hk-dh-chpsebceddr-novel-infectious-agent>

Comment 2.16. 1.252: could the estimated infection curve be shown?

Response 2.16. The estimated infection curve of local cases is now shown as the dark orange curve in Figure 1B.

Supplementary information

Comment 2.17. 1.15/16: "we used a deconvolution-based method" needs more detail and equations and/or code

Response 2.17. We did not include the MATLAB codes in the original submission because we used a standard deconvolution method which was commonly used in AIDS and influenza research (Becker et al, Stat Med., 1991, Goldstein et al, PNAS, 2009 and Wu et al, PLoS Med., 2011). The method from Becker et al is implemented in the "backprojNP" function within the surveillance R package. We used the same method as Goldstein et al and have included the self-defined MATLAB function "DeconvolutionIncidence1.m" in the revised submission.

Comment 2.18. 1.21: mean and SD have uncertainty in ref. 3 - what is the rationale for ignoring this, and what are the consequences on results? There seems to be a risk of spurious precision from ignoring this

Response 2.18. We acknowledge that there may be indeed be such a risk. However, the deconvolution method was only used to reconstruct the epidemic curve for R_t estimation with "EpiEstim" to look for suitable candidate proxies for mobility (Figure 2) and to compare with the R_t estimates from our own model built with mobility data (Figure 3). Our model was directly fitted to the observed dates of symptom onset of local cases in Hong Kong (see the Methods section), although we assumed the same incubation period distribution (mean 6.5 days and SD 2.6 days) when generating the time series of symptom onsets

from the time series of infections. Therefore any such potential risk, even if realized, would not have affected our findings or conclusions substantively.

Comment 2.19. 1.26: is the Gamma distribution a good choice given that the times are discrete? How was it converted into discrete counts of days?

Response 2.19. We used the equation for $f_{infection-reporting}$ to show the deconvolution assumed the incubation period distribution and the onset-to-reporting delay distribution are independent. However, when the deconvolution was applied, we used the cumulative density function for the time between infection and reporting, i.e.:

$$f_{infection-reporting}(t) = F_{infection-reporting}(t) - F_{infection-reporting}(t - 1)$$

Comment 2.20. 1.28: presumably the issue of missing uncertainty is also affecting the estimates of the delay from onset to reporting

Response 2.20. Please refer to our response to **Comment 2.18**.

Comment 2.21. 1.33: when using the Thompson et al. method, what prior was used, what generation time interval distribution and what time window, and why?

Response 2.21. We used the default prior in the “EpiEstim” R package, which in turn assumes a prior distribution for the effective reproduction number with mean of 5 and SD of 5. The generation time distribution was jointly estimated from the inference for R_t with symptom onset dates from 50 infector-infectee pairs collected in mainland China and cases exported outside mainland China during the early stage of the pandemic (from Leung et al, The Lancet, 2020). We assumed a 7-day time window in R_t estimation with “EpiEstim”, because the Octopus transport data showed obvious weekly patterns with transport volume over weekends was on average 60-70% of the weekdays of the same week and the onset-to-reporting delay was about 4 days throughout the second wave in Hong Kong. We have revised the Methods section accordingly.

Comment 2.22. 1.49: the model equations are missing some definitions, e.g. what is π_a , what is $I_b(t, \tau)$? All symbols used should be introduced and not left for the reader to guess

Response 2.22. $\pi_a(t)$ was the force of infection of age group a at time t which is defined using the equation

$$\pi_a(t) = \frac{1}{N(t)} \sum_{b=1}^m \int_0^t \beta_{ab}(t) I_b(t, \tau) d\tau$$

τ was a dummy variable used in the integration equation, m was the number of age groups in the population, and $I_b(t, \tau)$ was the number of infectious individuals in age group b at time t who got infected before time t . We have revised the Methods section accordingly.

Comment 2.23. 1.53/54: "pdf of the generation time was assumed to be Gamma and the same as that of the serial interval" - this is known to be wrong (see, e.g., Ganyani et al., 2020); what are the consequences of this assumption?

Response 2.23. We agree with the reviewer that generation time is not the same as the serial interval, and we have revised the Methods section accordingly to clarify that we did not mean they are the same but only assumed the two distributions have the same probability density function. Please see our response to **Comment 2.24.** for details.

Based on the onset dates of 50 infector-infectee pairs collected in mainland China and cases exported outside mainland China during the early stage of the pandemic, the estimated mean and SD of the generation time distribution from the joint inference are 4.74 (4.33-5.21) and 2.37 (2.17-2.61) days respectively (Figure S3). Our estimate of generation time is consistent with the estimates of generation time in Singapore by Ganyani et al (5.20 (3.78-6.78) days) but is slightly longer than that of Tianjin (3.95 (3.01-4.91) days). However, the outbreak in Tianjin was mainly driven by several large clusters (e.g. out of 135 confirmed cases, there were 45 confirmed cases from Baodi shopping centre cluster) and the observed serial intervals might have been shortened due to the non-pharmaceutical interventions during the nationwide lockdown period (Ali et al, Science, 2020).

In the revised submission, we decided not to include the inference of generation time distribution in the model built with Octopus data. Instead, we assumed that the generation time distribution was gamma with (i) mean 5.2 days and coefficient of variation 0.33 in the base case; and (ii) mean 4.2 and 6.2 days with the same coefficient of variation in the sensitivity analysis. We have revised the main text and the Methods section accordingly.

Comment 2.24. 1.73/74: also seems to confuse the serial and generation interval which, I think, are important to separate even if assumed equal

Response 2.24. We agree with the reviewer and have revised the Methods section accordingly. In the revised submission, we decided not to include the inference of generation time distribution in the model

built with Octopus data. Instead, we assumed that the generation time distribution was gamma with (i) mean 5.2 days and coefficient of variation 0.33 in the base case; and (ii) mean 4.2 and 6.2 days with the same coefficient of variation in the sensitivity analysis.

Comment 2.25. 1.109: where do asymptomatic infections feature and are they assumed to have the same generation interval as symptomatic cases on which the estimates were based? Also, is there a chance that the generation interval changed over the course of the epidemic (e.g., due to control measures) and what are the consequences of this?

Response 2.25. We treated asymptomatic cases similarly as symptomatic cases in the analyses, thus asymptomatic cases were assumed to have the same generation time distribution as symptomatic cases. We did not assume any changes in generation time distribution over the course of the epidemic due to control measures, because no stringent control measures similar to “lockdown” or “shelter-in-place” had ever been implemented in Hong Kong.

We used the case data from the official website of Centre for Health Protection in Hong Kong and cases were labeled as “asymptomatic” if they did not show any symptoms when they were confirmed by RT-PCR. Some cases might have shown symptoms during their hospitalizations after they were confirmed and admitted to hospitals, but this information was missing from the data we used. Specifically, as discussed in our response to **Comment 2.13.**, although some SARS-CoV-2 infections were asymptomatic at the time of reporting, for simplicity we used the term “case” here to include all symptomatic and asymptomatic SARS-CoV-2 infections.

Comment 2.26. Table S2/S3: are these missing uncertainty? Is this uncertainty reflected in the model?

Response 2.26. Table S2 and S3 showed the mean reported number of contacts from our previous POLYMOD-like contact survey in Hong Kong (Leung et al, Scientific Reports, 2017). The survey data are available on <http://www.socialcontactdata.org/>.

The survey data were not used in the inference but were presented for easier comparison to the estimated contact matrix from the model inference (Table S4).

Comment 2.27. Fig S1 is hard to assess visually - perhaps compare the R_t estimates with a scatter plot? R_t estimates are shown with uncertainty but since this is turned into a single value for the Pearson's correlation coefficient, would it make sense to just show this aggregate value here?

Response 2.27. We did not calculate Pearson's correlation coefficients for Figure S1. We simply used Figure S1 to show how accurate R_t could be estimated from deconvoluted epidemic curves of reported cases. The epidemic scenarios were simulated with pre-specified R_t (as shown as the blue lines in Column 2-4 of Figure S1). Please see details in our response to **Comment 2.7**.

Comment 2.28. Fig. S3: the posterior distributions look reassuringly similar but it would be good to also show parameter traces to assure the reader that convergence was obtained in the MCMC

Response 2.28. We have added MCMC trace plots accordingly in Figure S7 in the Supplementary Information.

REVIEWER COMMENTS

Reviewer #2 (Remarks to the Author):

The authors did a veru thorough job at revising the manuscript, which I found much improved. I have the following remaining comments/concerns.

- "Thus, we have only two models in the manuscript: 1) the model used by the "EpiEstim" package; and [...]. We did not include any renewal equation model throughout the analysis". I don't think this is correct: EpiEstim is a software package to estimate R using a discrete renewal equation model.

- l.123-125: "all of the epidemic forecasts from our framework converged with the observed epidemic curve during February-April (Figure 3)". As far as I can see, incidence is systematically overestimated in the ascending phase and underestimated in the descending phase. The sentence should be changed to clarify the relationship between forecasts and data ideally using some sensible error metric. This can be done even at small numbers, independently of the likelihood function used - I am afraid I do not follow the comment on Poisson likelihood in the response to reviewers. Since predictive accuracy is such a central claim of the paper I think that pointing readers to a visual assessment is not enough. In particular, the fourth forecast point seems key as it predicts a turnover, which models wihtout using a predictor such as mobility would not be able to predict, but then ends up underestimating cases - could this point to a systematic issue? As a reader I was left excited about the potential shown here for the use of digital proxies to improve forecasts but unconvinced that this was really possible without a more thorough investigation.

Thank you for sharing more of the code - as a minor comment this could do with more documentation for readability (e.g. it took me quite a while to work out the deconvolution code).

Point-by-point response to comments for “Real-time tracking and prediction of COVID-19 infection using digital proxies of population mobility and mixing”

Comments from Reviewer #2 (Remarks to the Author):

Comment 2.1. The authors did a very thorough job at revising the manuscript, which I found much improved. I have the following remaining comments/concerns.

Response 2.1. Thank you for the supportive comments. Please see below for our point-by-point response.

Comment 2.2. "Thus, we have only two models in the manuscript: 1) the model used by the “EpiEstim” package; and [...]. We did not include any renewal equation model throughout the analysis". I don't think this is correct: EpiEstim is a software package to estimate R using a discrete renewal equation model.

Response 2.2. Thank you for the comment and we apologize for the confusion. What we meant was we did not build any additional renewal equation model on our own other than using the model built in the EpiEstim package. We have revised the Methods section accordingly:

“Our analysis comprised the following steps:

1. Estimate the instantaneous reproductive number R_t of local cases in Hong Kong on each day t between 22 January (day 0) and 15 May (day T) **in the renewal equation model built in the “EpiEstim” package.**”

Comment 2.3. l.123-125: "all of the epidemic forecasts from our framework converged with the observed epidemic curve during February-April (Figure 3)". As far as I can see, incidence is systematically overestimated in the ascending phase and underestimated in the descending phase. The sentence should be changed to clarify the relationship between forecasts and data ideally using some sensible error metric. This can be done even at small numbers, independently of the likelihood function used - I am afraid I do not follow the comment on Poisson likelihood in the response to reviewers. Since predictive accuracy is such a central claim of the paper, I think that pointing readers to a visual assessment is not enough. In particular, the fourth forecast point seems key as it predicts a turnover, which models without using a predictor such as mobility would not be able to predict, but then ends up underestimating cases - could this point to a systematic issue? As a reader I was left excited about the potential shown here for the use of digital proxies to improve forecasts but unconvinced that this was really possible without a more thorough investigation.

Response 2.3. Thanks for the comments. In the revised manuscript, we used six metrics proposed in Funk et al to assess the prediction errors in our framework (please see the figure below for the results). Based on these assessment results, we have revised the relevant section on nowcasting and forecasting in Results as follows.

“We used the six metrics proposed in Funk et al (namely calibration, sharpness, bias, ranked probability score (RPS), Dawid-Sebastiani score (DSS) and absolute error (AE)) to assess the predictive performance of our forecasts (Figure S7)¹⁹. Sharpness, RPS, DSS and AE dropped progressively between 30 January and 28 February, suggesting that the performance of our forecast model was improving as more mobility and case data became available (Figure S7). However, all six metrics suggested that the forecasts were less accurate on 29 February, 15-17 March and 22-23 March (in the sense that the observed case counts were near or outside the 95% prediction intervals). These prediction errors were likely due to the occurrence of superspreading events, which tend to result in more explosive growth in incidence²⁰, and the stochastic effects in an otherwise low-prevalence setting. Specifically, our forecast underestimated the number of new onsets on 29 February which comprised a cluster from a religious group and another cluster seeded by returnees on the Diamond Princess Cruise. Similarly, our forecast underestimated incidence on 22-23 March which comprised a large cluster from a bar setting¹⁷. Because SSEs were rare in the training data and not explicitly modelled in our framework, the forecast model underestimated incidence when SSEs occurred. On the other hand, our forecast overestimated incidence on 15-17 March probably because the true prevalence was very low (on the order of tens) and the associated stochasticity in transmission dynamics resulted in an epidemic take-off that occurred slower than predicted by our deterministic framework. Notwithstanding the prediction errors attributed to SSEs and very low prevalence, our framework generated largely robust epidemic nowcasts during February-April: the estimated number of cases and its 95% prediction intervals from the fitted model (which were sufficiently tight for practical purposes) provided a reasonably robust inference of the number of cases that had already been generated but not yet registered by surveillance due to the infection-to-reporting delay (Figure 3). Under the assumption that population mixing would remain at status quo for 6 days following the time of prediction, our epidemic forecasts were congruent with the observed epidemic curve during February-April except when SSEs occurred or when prevalence was very low (Figure 3).”

Furthermore, we have added the following paragraph in the Discussion to discuss the effects of superspreading events on the performance of our framework:

“Fourth, our framework is based on changes in population mixing and therefore might not be able to provide accurate epidemic forecast when there are not enough data from large clusters or superspreading events to train the model (Figure 3). Changes in population mixing correlate with the average disease transmissibility well, but the occurrence and scale of superspreading events are often

associated with substantial stochasticity. To improve our framework, estimation of the scale of superspreading events should be integrated into the model once the relevant data are available from outbreak investigations.”

Figure S7. Forecasting metrics and scores of the forecast performed 9 days ahead of the time point shown on the x-axis². (A) Calibration (p-value of Anderson-Darling test, dashed lines at 0.1 and 0.01); (B) Sharpness (values closer to 0 indicate sharper models); (C) Bias (values closer to 0 indicate less bias); (D) Ranked probability score (values closer to 0 indicate better forecast); (E) Dawid-Sebastiani score (values closer to 0 indicate better forecast); (F) Absolute error (values closer to 0 indicate better forecast). The light red shades indicated that the forecast model did not perform well on 29 February and 22-23 March due to the occurrence of large clusters or superspreading events.

Comment 2.4. Thank you for sharing more of the code - as a minor comment this could do with more documentation for readability (e.g. it took me quite a while to work out the deconvolution code).

Response 2.4. Thank you for your comment. The function “DeconvolutionIncidence1.m” follows the methods in Goldstein et al. (<https://www.pnas.org/content/106/51/21825>). This function inputs observed incidence (numbers of cases, deaths, hospitalizations, etc. per day), and uses an adaptation of Richardson-Lucy deconvolution to recover the underlying incidence curve. We have included more description in the “DeconvolutionIncidence1.m” file.

The methods have been used in the below paper by Gostic et al and we have added in the reference list of the manuscript: Gostic KM, McGough L, Baskerville SA, Joshi K, Tedijanto C, Kahn R, Niehus R, Hay J, de Salazar P, Hellewell J, Meakin S. Practical considerations for measuring the effective reproductive number, Rt. medRxiv. 2020 Jan 1.

We have also referenced the corresponding GitHub repository with a similar deconvolution function in R “Richardson_Lucy.R”: https://github.com/cobeylab/Rt_estimation/blob/master/code/Richardson_Lucy.R

REVIEWERS' COMMENTS

Reviewer #2 (Remarks to the Author):

The authors have done a great job with the revisions - I think the paper is much improved and am happy to recommend acceptance.

I just have one last question: I wonder if Fig. S7 is a misunderstanding of "calibration" which can really only be assessed across a whole time series of forecasts rather than at each time point (as otherwise the test has no power). Can I suggest that you remove this panel and instead you just report overall (across the whole time series) coverage at e.g. the 50 and 90% level across the time series, such as e.g. here:

<https://www.medrxiv.org/content/10.1101/2020.08.19.20177493v1>

That is, just report the proportion of data points that fall within the 50%/90% predictive interval across the whole time series.

Point-by-point response to comments for “Real-time tracking and prediction of COVID-19 infection using digital proxies of population mobility and mixing”

Comments from Reviewer #2 (Remarks to the Author):

Comment 2.1. The authors have done a great job with the revisions - I think the paper is much improved and am happy to recommend acceptance.

Response 2.1. Thank you for the supportive comments. Please see below for our point-by-point response.

Comment 2.2. I just have one last question: I wonder if Fig. S7 is a misunderstanding of "calibration" which can really only be assessed across a whole time series of forecasts rather than at each time point (as otherwise the test has no power). Can I suggest that you remove this panel and instead you just report overall (across the whole time series) coverage at e.g., the 50 and 90% level across the time series, such as e.g., here: <https://www.medrxiv.org/content/10.1101/2020.08.19.20177493v1>

That is, just report the proportion of data points that fall within the 50%/90% predictive interval across the whole time series.

Response 2.1. Thank you for the comment. In the Figure S7 of last revision, we assessed the forecasts of 6 days (i.e., the period covered by the green line and shades in Figure 3) by the six metrics including “calibration” at each time point (i.e., the forecast of 6 days). To avoid any misunderstanding, we followed the reviewer’s advice and removed the Panel A from Figure S7. Instead, we added Figure S8 which showed the 50%, 90% and 95% prediction intervals from the fitted model.

We have revised the main text accordingly as follows:

“We used the metrics proposed in Funk et al (namely sharpness, bias, ranked probability score (RPS), Dawid-Sebastiani score (DSS) and absolute error (AE)) to assess the predictive performance of our forecasts (Figure S7). Sharpness, RPS, DSS and AE dropped progressively between 30 January and 28 February, suggesting that the performance of our forecast model was improving as more mobility and case data became available (Figure S7). However, all metrics suggested that the forecasts were less accurate on 29 February, 15-17 March, and 22-23 March (Figure 3 and Figure S8, in the sense that the observed case counts were near or outside the 95% prediction intervals).”

Figure S7. Forecasting metrics and scores of the forecast performed 9 days ahead of the time point shown on the x-axis ². (A) Sharpness (values closer to 0 indicate sharper models); (B) Bias (values closer to 0 indicate less bias); (C) Ranked probability score (values closer to 0 indicate better forecast); (D) Dawid-Sebastiani score (values closer to 0 indicate better forecast); (E) Absolute error (values closer to 0 indicate better forecast). The light red shades indicated that the forecast model did not perform well on 29 February and 22-23 March due to the occurrence of large clusters or superspreading events.

Figure S8. Retrospective nowcast and forecast of the COVID-19 epidemic in Hong Kong. The green dash lines indicate the dates on which retrospective nowcast, and 6-day forecast were made (2 March, 14 March, 17 March, 22 March and 4 April). The blue dash lines indicate the latest empirical R_t estimates obtainable from the epidemic curve on those dates. (A) Comparison between empirical R_t estimates and R_t from the fitted SIR model. The red line and shades indicate the empirical R_t estimates. Blue lines and shades indicate the nowcasted R_t . Green lines and shades correspond to the assumption that population mixing (and hence R_t) would remain at status quo for the next 6 days. (B) Nowcast and forecast of local epidemic curves by dates of symptom onset. Red bars showed the number of local onsets reported on or before the date of prediction. Orange bars showed the number of local onsets reported after the date of prediction. Blue lines and shades (i.e., showing 50%, 90% and 95% CrIs of posterior distribution from lighter to darker shades) indicate the nowcasted local epidemic curve. Green lines and shades (i.e., showing 50%, 90% and 95% CrIs of posterior distribution from lighter to darker shades) indicate the forecasted local epidemic curve.